# The Effectiveness of Physical Activity Interventions on Depression in Korea: A Systematic Review and Meta-Analysis

**DOI:** 10.3390/healthcare10101886

**Published:** 2022-09-27

**Authors:** Ye Hoon Lee, Hyungsook Kim, Heetae Cho

**Affiliations:** 1Division of Global Sport Industry, Hankuk University of Foreign Studies, Gyeonggi-do 17035, Korea; 2Department of Cognitive Science, School of Intelligence, Hanyang University, Seoul 04763, Korea; 3Department of Sport Science, Sungkyunkwan University, Gyeonggi-do 03063, Korea

**Keywords:** depression, exercise, mental health, physical activity, quality of life, randomized controlled trials

## Abstract

Since the onset of the COVID-19 pandemic, Korea has ranked first in the OECD, with a prevalence of 36.8% of depression. Thus, this study aimed to estimate the effect size of physical activity as an alternative tool for depression symptoms using meta-analysis. A meta-analysis on depressive symptoms was performed on 18 studies published in Korean domestic journals. The moderating variables hypothesized in this study included age groups of participants; depressive symptoms; and frequency, intensity, time, type, and duration of the intervention. The overall effect size of physical activity on depression was moderate (0.56 [95% CI: 0.39 to 0.91]). Specifically, physical activity was slightly more effective in reducing depression in participants with an 18–64 age group compared to older people over 65 years old, while it was most effective for participants without depressive symptoms compared to participants with mild and severe symptoms. Further, the subgroup analysis revealed that performing two times a week for 30 to 60 min with progressive intensity for 1–8 weeks may be the most effective for Koreans. The results of this study can provide guidelines for the most effective physical activity program for Koreans.

## 1. Introduction

Depression is a major psychiatric disorder affecting more than 300 million people worldwide, with a lifetime prevalence of 14.6% and a 1-year prevalence of 5.9% [1,2,3]. It has been found to cause an enormous socioeconomic burden, including low job performance [4,5]; poor health-related quality of life [6]; reduced cognitive abilities [7]; increased health care costs [8], and high mortality [9]. Additionally, a previous review reported that approximately 60% of all suicides were associated with depression, making depression one of the most important risk factors for suicide [10].

In Korea, the recent increased number of people with depression is alarming as the number has increased by more than 30% in the past five years [11], while the rate of suicidal ideation also increased from 9.7% in 2020 to 16.3% in 2021 [12]. In fact, the prevalence of depression in Korea in 2020 was 36.8%, ranking first among the Organization for Economic Cooperation and Development (OECD) countries [13]. However, Korea has been known as one of the most difficult countries to receive treatment for depression because of the 60-day prescription limit for safe selective serotonin reuptake inhibitor (SSRI) antidepressants announced by the government in 2002. Due to this regulation, non-psychiatric doctors, who accounted for 96% of all doctors, were suddenly unable to treat depression [14]. Subsequently, the suicide rate in Korea reached its highest among OECD countries [15]. With the limited accessibility of depression management, the early detection and management of depression and the development of self-intervention to prevent depression are warranted in Korea.

In the treatment of mental disorders, such as depression, physical activity is increasingly recognized as a means of improving mental health, minimizing side effects, and reducing depression [16], as there are currently no breakthrough treatments that have fewer side effects and can reduce depressive symptoms [17]. Indeed, empirical evidence, along with the results from systematic reviews and meta-analyses, has also reported that physical activity has the potential to reduce depressive symptoms with a moderate-to-large effect size and can be a useful addition to pharmacotherapy and psychotherapy [18,19,20,21,22] by increasing the levels of serotonin, dopamine, and norepinephrine [23,24,25] and improving psychological factors, such as self-esteem and self-efficacy [26].

Studies on the effect of physical activity on depression have been conducted steadily in Korea, and in general, physical activity has been reported to have significant effects on depression. For example, Kim and Kim [27] presented a mixed exercise that combined aerobic exercise and resistance exercise two–three times a week for eight weeks that demonstrated a significant effect on depression. However, in terms of improving mental health, the recommended FITT (frequency, intensity, time, and type) principle [28] should consider the individual’s depressive symptoms, age, gender, and nationality in order to treat, observe, and promote the physical activity of various people or groups [29]. Furthermore, a meta-regression analysis conducted by Bellon and colleagues [30] revealed lower effectiveness of physical activity in Asia when compared with countries on other continents, such as in North America, Europe, and South America. Therefore, to solve the optimal FITT principles for depression and thereby reduce depression among Koreans, it is necessary to comprehensively analyze the results of individual studies through a systematic method.

Against this background, the purpose of this study was to apply a systematic review and meta-analysis (1) to understand the effects of the physical activity on depression in randomized controlled experimental study in Korea and (2) to determine the effect of physical activity on depression according to exercise type, exercise time, exercise intensity, exercise period, and exercise frequency.

## 2. Materials and Methods

### 2.1. Trial Eligibility Criteria

Based on the Preferred Reporting Items for Systematic Review and Meta-analysis (PRISMA), we used the PICOS (Population-Intervention-Comparison-Outcome [s]-Study design) framework for the eligibility criteria for this meta-analysis [31]. More specifically, to obtain a sufficient number of studies to conduct the analysis, the eligible study subjects (P) were persons of all ages who participated in any type of physical activity; the eligible intervention methods (I) were all types of physical activity, which is defined as “any bodily movement produced by the skeletal muscles that results in energy expenditure above resting levels” [31]; the comparative group (C) was a wait-list control group that did not participate in any type of physical activity. For the study result (O), studies were included when they used a previously validated depression symptom scale (i.e., CES-D, BDI, GDS for older people, etc.) and selected the depression symptoms as dependent variables. For the study design type (S), only randomized controlled studies were included in this meta-analysis. In the case of a study in which other conditions were appropriate for meta-analysis but data were not sufficiently presented, the investigator was contacted directly to obtain supplementary data.

The exclusion criteria for data analysis are as follows: (1) non-experimental studies (e.g., survey studies, qualitative studies, case studies, etc.); (2) single experimental designs without a control group; (3) studies not consistent with PICOS; (4) duplicate publications reporting the same results; (5) gray papers not peer-reviewed (abstract, poster); (6) non-Korean Studies; and (7) research in which in which statistical values are missing.

### 2.2. Data Source

We used electronic databases in Korea, including the Korea Education and Research Information Service (KERIS), National Assembly Library (NAL), Research Information Service System (RISS), and Korean Studies Information Service System (KISS). We used broad search terms in order to capture a wide spectrum of studies, specifically the following search algorithm (Table A1): “physical activity” OR “exercise” OR “sport” OR “leisure” OR “physical exertion” OR “aerobic” OR “resistance” OR “mixed” OR “mind-body” OR “yoga” OR “running” OR “walking” OR “bicycling” AND “depress” OR “depression” OR “depressive disorder”.

### 2.3. Data Extraction and Coing

Studies satisfying all of the above inclusion criteria were classified by combining the research author and publication year and coded as follows: participant characteristics (number of participants, average age, gender of the participants, depression severity, other symptoms), physical activity characteristics (frequency, intensity, time, type, and duration), situational characteristics (individual/group, time period of the physical activity (daytime/evening), and research characteristics (author, publication year, type of experience, and the used survey).

### 2.4. Evaluation of the Risk of Bias

To evaluate the risk of bias in the selected literature, the revised Cochrane risk-of-bias tool for randomized trials 2 (RoB2) in 2019 was used [32]. RoB2 was completed in five domains as an evaluation tool for randomized controlled trials (RCT). The detailed evaluation details are: (1) bias arising from the randomization process; (2) bias due to deviations from intended interventions; (3) bias due to missing outcome data; (4) bias in the measurement of the outcome; and (5) bias in the selection of the reported result. In this order, the detailed questions in each area were answered with “Yes (Y)” OR “Probably yes (PY)”, “No (N)” OR “Probably no (PN)”, OR “No information (NI)”. Finally, using the responses of each item, the risk of bias was evaluated as “low risk of bias”, “some concerns”, and “high risk”, according to the algorithm provided by RoB2.2.4. Statistical analysis

#### 2.4.1. Measure of Effect

In the process of verifying the effect size, a random-effects model was selected, and the average effect size for each study (or the overall average effect size) was calculated. As all data were continuous data, the effect size of each study was calculated as standardized mean difference (SMD). The effect size of physical activity for measures of depression symptoms was calculated as the pre-post difference (within-group) for each study. For all effect sizes, Hedges’ g was used to correct for bias owing to the small sample size [33]. Hedges’ g is interpreted as small (0.2), medium (0.5), or large (0.8) effect size.

#### 2.4.2. Heterogeneity

First, Q, the observed variance of each effect size—that is, the total variance including both the sampling error variance and the actual inter-study variance—is used to verify the homogeneity of the effect size. Second, the researchers also used a heterogeneity test when the homogeneity test was rejected, that is, when the distribution of effect sizes was heterogeneous. Usually, *T*^2^, which represents the inter-study variance, or *I*^2^, which represents the actual variance ratio, is used. The formula for calculating *I*^2^ is as follows: in addition, when *I*^2^ is at 25%, it can be judged that there is small-scale heterogeneity; when 50% there is medium-sized heterogeneity; and when *I*^2^ is at 75%, there is large-scale heterogeneity [34].

#### 2.4.3. Publication Bias

Publication bias refers to a problem in the representativeness of sampling by analyzing only some papers without analyzing all studies related to the research topic [34]. To verify this, forest and funnel plots were used. The forest plot is the most basic result of a meta-analysis and includes the average effect sizes of individual studies, the average effect sizes of all studies, statistical significance, and weight. In the case of a funnel plot, if the plot is symmetrical, it is considered that there is no publication bias because it is the same as the result assuming that all subjects were included in the study [33].

#### 2.4.4. Subgroup Analysis

Next, moderation subgroup analysis was used to examine the factors that cause heterogeneity when it was determined that heterogeneity existws. Meta-ANOVA and meta-regression are available as moderation effect analysis methods that can directly verify the differences in effect sizes between subgroups. In this study, we used meta-ANOVA, as the moderator variables were categorical variables [34]. Specifically, we used several moderators, including (1) participant characteristics (age and depression symptoms) and (2) physical activity characteristics (frequency, intensity, time, type, and duration).

## 3. Results

### 3.1. Characterisitcs of Included Studies

In this study, 18 research articles were selected and used in the meta-analysis as final analysis targets based on the PICOS selection criteria (Figure 1). Table 1 presents the characteristics of the final analysis.

### 3.2. Homogeneity Test

As a result of calculating the homogeneity test to see whether 18 studies were based on a homogeneous population, the homogeneity test statistic Q was 84.096, which was statistically significant (*p* = 0.000), indicating that the 18 studies were not based on the same population. As a result of the heterogeneity analysis, *T*^2^, the variance between studies, was 0.44, and *I*^2^, the actual degree of variance, was 79.8%, which can be interpreted as reflecting a large degree of heterogeneity in the 18 studies [35]. Figure 2 shows the effect sizes of the 18 studies in a forest plot.

### 3.3. Risk of Bias

The results of the risk of bias evaluated according to the algorithm of the ROB 2.0 tool for 18 selected studies are as follows. In the 18 RCT studies, 15 papers (83.3%) applied a randomization process and allocation concealment, so their final risk was judged to be low. Next, looking at the bias that deviated from the intended intervention and bias in outcome measurement, 11 studies (61.1%) blinded participants, while 13 studies (72.2%) blinded the outcome evaluation. In addition, bias due to missing data was judged as low risk because the 18 studies had low dropout rates and almost no data loss.

Additionally, looking at the error in outcome measurement, only 10 of the 21 studies performed evaluator blinding, but the evaluation was not likely to be affected by the knowledge of the intervention, and the outcome measurement method was appropriate, so these studies were judged as low risk. Finally, all 18 cases were judged as low risk because no particular bias could be found in the selection of the results for errors due to the selected report.

### 3.4. Effectiveness of Physical Activity Interventions

As shown in Table 2 and Figure 2, the effect sizes for the 18 studies were calculated for depression, and the weighted average summary effect size was 0.555 (*p* < 0.001), which could be interpreted as statistically significant and moderate (ES > 0.5). As a result of testing homogeneity, *I*^2^ was 79.8%, indicating that the heterogeneity of the effect size was considerable (Q = 84.09; *p* = 0.000). In other words, the study characteristics were heterogeneous. Therefore, subgroup analyses were performed to determine the causes of the heterogeneity between studies.

### 3.5. Publication Bias

To verify the validity of the results, a publication bias test was conducted. A funnel plot is a scatter plot drawn using the effect size on the *X*-axis and the number of samples or standard errors on the *Y*-axis. If there was no publication bias in the meta-analysis results, the plots appeared in a symmetrical funnel shape. As a result of confirming the publication bias funnel plot in Figure 3, the scatterplot was symmetrical, so it can be interpreted that there was no publication bias. As a result of verifying publication bias through Egger’s regression intercept significance test, the regression intercept was 3.22, the standard error was 1.96, and *p* = 0.120 (two-tailed), indicating that there was no publication bias.

### 3.6. Subgroup Analysis and Meta-Regression

From Table 2, it is possible to confirm the effect size and heterogeneity between studies according to the moderating variables of the study on depressive symptoms.

#### 3.6.1. Age Group

For the age groups of participants, it was found that the effect size of the study was larger for adults (SMD = 0.68; 95% CI 0.20 to 1.17; age > 18, <64) compared with seniors (SMD = 0.65; 95% CI −0.11 to 1.41; age > 65) and adolescents (SMD = −0.86; 95% CI −1.64 to −0.10; age < 18). Moderator analysis revealed that heterogeneity was statistically significant (Q_2_ = 12.12, *p* = 0.002; *I*^2^ = 84.39%).

#### 3.6.2. Depression Severity

A high effect size was found for studies with normal participants (SMD = 1.011; 95% CI 0.35 to 1.67; Q_5_ = 25.80, *p* = 0.000; *I*^2^ = 81.9%), whereas a low effect size was found for studies with participants with mild (SMD = 0.39; 95% CI −0.23 to 1.02; Q_5_ = 19.94, *p* = 0.001; *I*^2^ = 77.95%) and severe (SMD = 0.27; 95% CI −0.41, 0.97; Q_5_ = 25.36, *p* = 0.000; *I*^2^ = 85.87%) symptoms. There were no statistically significant differences between the subgroups (Q_3_ = 5.44, *p* = 0.142; *I*^2^ = 83.63%).

#### 3.6.3. Frequency

A high effect size was found for studies with activity two times a week (SMD = 0.72; 95% CI 0.12 1.32.; Q_7_ = 39.20, *p* = 0.000; *I*^2^ = 82.54%), while a low effect size was found for studies with three times a week (SMD = 0.48; 95% CI −0.11 to 1.06; Q_8_ = 40.22, *p* = 0.000; *I*^2^ = 85.85%) and seven times a week (SMD = 0.27; 95% CI −0.41 to 0.97; Q_5_ = 25.36, *p* = 0.000; *I*^2^ = 85.87%). Thus, when the participants engaged in physical activity twice a week, the effect size of the study was larger than those with three times a week or seven times a week, but the difference was not statistically significant (Q_2_ = 2.47, *p* = 0.291; *I*^2^ = 83.63%).

#### 3.6.4. Intensity

A high effect size was found for studies with progressive intensity (SMD = 1.93; 95% CI 1.01 to 2.85; Q_0_ = 39.20, *p* = 0.000; *I*^2^ = 82.54%), followed by moderate-vigorous (SMD = 0.67; 95% CI −0.43 to 1.78.; Q_2_ = 11.83, *p* = 0.003; *I*^2^ = 84.99%), light (SMD = 0.66; 95% CI −0.04 to 1.35.; Q_4_ = 24.51, *p* = 0.000; *I*^2^ = 86.66%), and moderate (SMD = 0.55; 95% CI −0.68 to 1.78.; Q_3_ = 32.92, *p* = 0.003; *I*^2^ = 84.99%) intensity, but the difference was not statistically significant (Q_3_ = 5.66, *p* = 0.129; *I*^2^ = 87.78%).

#### 3.6.5. Duration

A large effect size was found for studies that required 1 to 8 weeks (SMD = 0.73; 95% CI 0.21 to 1.26; Q_6_ = 26.44, *p* = 0.000; *I*^2^ = 79.51%), while a small effect size was found for studies that required 9 to 16 weeks (SMD = 0.29; 95% CI −0.22 to 0.80); Q_9_ = 35.58, *p* = 0.000; *I*^2^ = 81.58%), and this difference was statistically significant. (Q_2_ = 11.48, *p* = 0.003; *I*^2^ = 83.63%)

#### 3.6.6. Time

A high effect size was found for studies that required 31 to 60 min per session (SMD = 0.77; 95% CI 0.01 to 1.52.; Q_7_ = 47.25, *p* = 0.000; *I*^2^ = 86.03%), while a small effect size was found for studies that required 1 to 30 min per session (SMD = 0.31; 95% CI −0.02 to 0.60; Q_3_ = 2.53, *p* = 0.470; *I*^2^ = 00.00%). The subgroup analysis indicated that there was no statistically significant difference between the variables (Q_2_ = 4.31, *p* = 0.116; *I*^2^ = 83.25%).

#### 3.6.7. Type

A large effect size was found for studies with mixed types of exercise (SMD = 0.79; 95% CI −0.34 to 1.92; Q_4_ = 40.20, *p* = 0.000; *I*^2^ = 90.56%), followed by aerobic exercise (SMD = 0.65; 95% CI 0.04 to 1.26; Q_4_ = 13,93, *p* = 0.008; *I*^2^ = 73.34%) and mind-body exercise (SMD = 0.53; 95% CI −0.44 to 1.49; Q_3_ = 18.28, *p* = 0.000; *I*^2^ = 84.48%). Finally, a small effect size was found for studies with resistance training (SMD = 0.23; 95% CI −0.21 to 0.66; Q_2_ = 3.38, *p* = 0.184; *I*^2^ = 40.43%). However, this difference was not statistically significant (Q_3_ = 1.78, *p* = 0.619; *I*^2^ = 83.12%).

## 4. Discussion

This study was conducted to establish a basis for physical activity intervention guidelines for Koreans by confirming its effects. A meta-analysis of 18 selected articles was conducted through a systematic literature review of domestic randomized controlled experimental studies to determine the effects on depression. The discussion based on the results ia as follows.

First, the effect size of physical activity was moderate, whereas the heterogeneity between studies was found to be large. These results suggest that physical activity can be an alternative tool to reduce the frequency of drug use due to depression, improve independence in daily activities, and contribute to a healthy lifestyle [16,36]. This is a similar result to the moderate effect size in existing meta-analyses of the effectiveness of physical activity [37,38]. However, the effect size in this study was also lower than that in the study by Bellon and colleagues [30] (SMD = −0.34). This difference may be attributed to the characteristics of the population. That is, the current study included studies that used participants with various depressive symptoms ranging from normal to severe for the subgroup analysis, whereas Bellon et al. [30] included participants without clinical depression. Thus, the difference may indicate that the effectiveness of physical activity on depression can be stronger for those with certain levels of depressive symptoms (e.g., moderate to severe).

Further, as a result of analyzing the effect size for the reduction of depression according to FITT principles of physical activity, the subgroup analysis revealed that performing activity 2 times a week for 30 to 60 min (60–120 min per week) with progressive intensity for 1–8 weeks may be most effective for Koreans. For the past several decades, many Western countries and credible organizations have developed and provided guidelines for physical activity to the public for reasons such as the health benefits and economic effects of physical activity. In the West, in the 1990s, the American College of Sports Medicine (ACSM) and the Centers for Disease Control and Prevention (CDC) developed guidelines and recommended performing moderate-intensity aerobic exercise 5 days a week for 30 min or more at a time [39]. Recently, the WHO [40] summarized the amounts of physical activity required by age, pregnancy, chronic disease, or disability. For example, an 18- to 64-year-old woman without medical problems should receive at least 150–300 min of moderate-intensity aerobic exercise or 75–150 min of vigorous exercise per week. It is also recommended to perform muscle-strengthening exercises that involve all major muscle groups at least two days a week. This study has significant implications by demonstrating specific guidelines of physical activity for Koreans, for whom there is insufficient evidence on the specific FITT principle of the effectiveness of physical activity. Thus, this study is meaningful in that it provides the basis for the best physical activity for Koreans, as the types of physical activity were more systematically classified, according to the guidelines and presented the effect size of each moderating variable.

This study has implications for the effective implementation of physical activity by instructors in the field of fitness and exercise by examining the moderating variables for the size of the effect of physical activity and revealing at which age level and for which duration physical activity is more effective. For example, various physical activity programs are being implemented on this site. Professional researchers in the field of physical activity as well as the government assert that it is important to reduce depression and recognize physical activity as an essential and professional treatment for improving depression. Therefore, based on the results of this study, it is expected that a more systematic and effective exercise program for depression will be developed and that the studied physical activity program will not only be academically verified but will actually contribute to the reduction of depression in Koreans.

### Limitation and Future Research Directions

The limitations of this study are as follows: First, although it was confirmed that mixed physical activity exercise was the most effective among the various types, there is a limit to confirming this result, as there are various specific types of mixed exercise. To confirm which exercise and method were more effective in mixed exercise in reducing depression, the recognition of specific types of mixed physical activity is suggested through a meta-analysis of functional effects. Further, to confirm the relationship between physical activity duration and depression, we suggest a long-term follow-up study to confirm the changes in depression according to increases or decreases in the amount of physical activity and maintaining the same amount of physical activity with the same physical activity type.

Second, there is concern about language bias by limiting the language to Korean. As this is a study on the effects of physical activity utilized in Korea, only previous studies related to the Korean population were analyzed. In a follow-up study, it is necessary to perform a meta-analysis of the effects of overseas physical activity and to compare and analyze the commonalities and differences between the effects of Korean and overseas physical activity. Similarly, this study only used previous research articles published in Korean domestic journals. A future meta-analysis could include research articles published in the international journals to obtain more relevant findings.

Further, it is important to note that the studies used in this meta-analysis did not use the same scale (e.g., BDI, CES-D, GDS, etc.), which can generate a potential risk in defining depression. Additionally, caution is required in generalizing the findings of this study as some groups had very few studies, which may affect the validity of the interpretation.

Finally, if previous studies related to the effect of physical activity conducted in a non-face-to-face online environment are accumulated due to the COVID-19 crisis that started in early 2020, the effect of physical activity conducted in the new training ecosystem will be determined by participant characteristics and physical activity characteristics. It is necessary to compare and analyze how effects differ by the extent to which participants engaged in physical activity in face-to-face or online contexts.

## 5. Conclusions

Korea has been ranked as having the highest prevalence of depression and suicide among OECD countries. This number has worsened during the recent COVID-19 period. As there is an urgent need for the early detection and management of such diseases, physical activity has been considered one of the solutions based on its merits in terms of accessibility and effectiveness. Although many studies have found the effectiveness of physical activity on depression, insufficient evidence has been demonstrated for which types of physical activity, along with the frequency, duration, and duration of physical activity, may be the most effective in the Korean population. When it was difficult to select various physical activities for Koreans, the effect was confirmed only by a randomized controlled experimental study with a high level of evidence in Korea. We hope that this study will make steady efforts to find countermeasures to reduce prevalence of depression in Korea.

## Figures and Tables

**Figure 1 healthcare-10-01886-f001:**
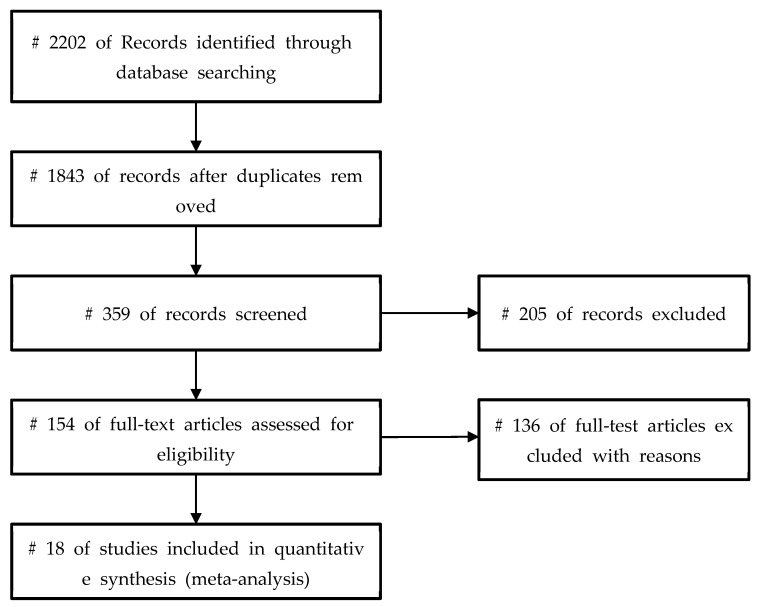
PRISMA flow diagram of the included study.

**Figure 2 healthcare-10-01886-f002:**
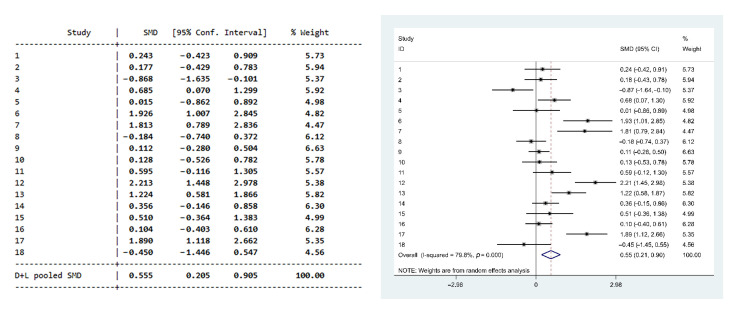
Forest plot of the overall mean effect size, as well as the effect size for each study included in the analysis.

**Figure 3 healthcare-10-01886-f003:**
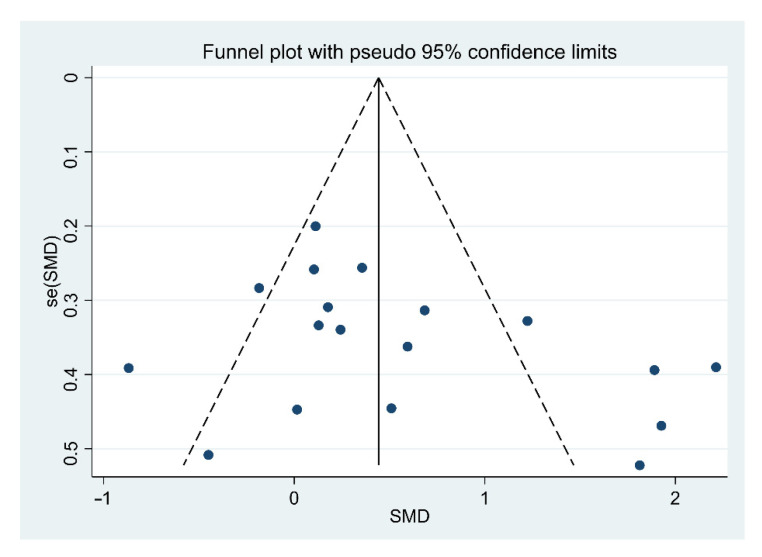
Funnel plot of the meta-analysis of the included research.

**Table 1 healthcare-10-01886-t001:** Characteristics of the included studies.

Research Characteristics				Participant Characteristics			PA Characteristics				
Author	Year	*N*		*M* Age		Severity	Frequency	Intensity	Time	Type	Duration
		I	C	I	C						
Kim	(1995)	17	18	50.17	50.17	Mild	3	-	-	Aerobic	6
Seo	(2003)	21	21	-	-	Severe	3	Mod	60	Aerobic	12
Lim et al.	(2016)	14	15	13.3	13.3	Severe	3	Mod	60	Mixed	15
Bang et al.	(2016)	18	27	42.22	37.37	Severe	2	Mod	10	Mixed	6
Lee	(2018)	10	10	68.5	68.5	Mild	2	-	30	Mixed	10
Jang et al.	(2015)	14	14	46.21	51.93	Severe	3	Progressive	60	Mixed	12
Kim & Kim	(2020)	11	11	65	65	Normal	3	Mod-Vig	60	Aerobic	12
So et al.	(2009)	25	25	23.22	25	Severe	2	Mod-Vig	75	Resistance	14
Kim et al.	(2013)	50	50	70.2	69.5	Mild	3	Low	40	-	10
So et al.	(2010)	18	18	63.1	65.4	Normal	2	Low	60	Aerobic	12
Son et al.	(2015)	16	16	72.2	73.8	Normal	2	Mod-Vig	40	Resistance	12
Choi	(2001)	23	21	43.52	43.95	Normal	2	Mod	40	Mixed	8
Moon	(2006)	27	19	-	-	Normal	3	Low	60	Aerobic	6
Ahn et al.	(2015)	31	31	54	53	Normal	3	-	20	Resistance	8
Ahn & Park	(2006)	11	10	34	32	Mild	2	-	35	Mind-body	6
Song & Park	(2001)	30	30	53	43.5	Severe	7	Low	20	Mind-body	2
Kim et al.	(2013)	17	22	79.5	75.9	Mild	2	Low	60	Mind-body	24
Jung et al.	(2016)	8	8	71.75	72.75	Mild	3	-	50	Mind-body	12

Note. I = Intervention group, C = Control group, Mod = moderate, mod-vig = Moderate to Vigorous. *M* age indicates participants’ mean age.

**Table 2 healthcare-10-01886-t002:** The results of the subgroup analysis.

Subgroup Analysis	K	SMD	95% CI	Heterogeneity	Test for Subgroup Difference
Age group					*X*^2^ = 12.12, df = 2, *p* = 0.002 **
Mean age < 18	1	−0.868	−1.635 to −0.101	*NA*	
Mean age > 18 < 65	10	0.682	0.200 to 1.165	*X*^2^ = 43.87, df = 9 (*p* = 0.000); *I*^2^ = 82.16%	
Mean age > 65	6	0.650	−0.105 to 1.405	*X*^2^ = 27.22, df = 5 (*p* = 0.000); *I*^2^ = 84.05%	
Depression symptom severity					*X*^2^ = 2.69, df = 2, *p* = 0.261
Mild	6	1.011	0.350 to 1.672	*X*^2^ = 25.80, df = 5 (*p* = 0.000); *I*^2^ = 81.93%	
Moderate	6	0.397	−0.225 to 1.020	*X*^2^ = 19.94, df = 5 (*p* = 0.001); *I*^2^ = 77.95%	
Severe	6	0.277	−0.415 to 0.969	*X*^2^ = 25.36, df = 5 (*p* = 0.002); *I*^2^ = 85.87%	
Frequency					*X*^2^ = 2.47, df = 2, *p* = 0.291
2	8	0.721	0.119 to 1.323	*X*^2^ = 39.20, df = 7 (*p* = 0.000); *I*^2^ = 82.54%	
3	9	0.477	−0.108 to 1.062	*X*^2^ = 40.22, df = 8 (*p* = 0.000); *I*^2^ = 85.85%	
7	1	−0.403	0.610	*NA*	
Intensity					*X*^2^ = 5.66, df = 3, *p* = 0.129
Low	5	0.655	−0.041 to 1.352	*X*^2^ = 24.51, df = 4 (*p* = 0.000); *I*^2^ = 86.66%	
Moderate	4	0.550	−0.687 to 1.786	*X*^2^ = 32.92, df = 3 (*p* = 0.000); *I*^2^ = 92.41%	
Moderate-Vigorous	3	0.674	−0.430 to 1.779	*X*^2^ = 11.83, df = 2 (*p* = 0.003); *I*^2^ = 84.99%	
Progressive	1	1.926	1.007 to 2.845	*NA*	
Time					*X*^2^ = 4.31, df = 2, *p* = 0.116
1–30 min	4	0.309	0.018 to 0.600	*X*^2^ = 2.532, df = 3 (*p* = 0.470); *I*^2^ = 0.00	
31–60 min	8	0.770	0.014 to 1.525	*X*^2^ = 47.25, df = 7 (*p* = 0.000); *I*^2^ = 86.03%	
61–90 min	1	−0.184	−0.740 to 0.372	*NA*	
Type					*X*^2^ = 1.78, df = 3, *p* = 0.619
Aerobic	5	0.655	0.047 to 1.264	*X*^2^ = 13.93, df = 4 (*p* = 0.008); *I*^2^ = 73.74%	
Resistance	3	0.228	−0.207 to 0.663	*X*^2^ = 3.380, df = 2 (*p* = 0.184); *I*^2^ = 40.43%	
Mind-body	4	0.527	−0.439 to 1.493	*X*^2^ = 18.28, df = 3 (*p* = 0.000); *I*^2^ = 84.48%	
Mixed	5	0.789	−0.340 to 1.918	*X*^2^ = 40.20, df = 4 (*p* = 0.000); *I*^2^ = 90.56%	
Duration					*X*^2^ = 11.48, df = 2, *p* = 0.003 **
1–8 weeks	7	0.738	0.210 to 1.266	*X*^2^ = 26.44, df = 6 (*p* = 0.000); *I*^2^ = 79.51%	
9–16 weeks	10	0.291	−0.220 to 0.801	*X*^2^ = 35.58, df = 9 (*p* = 0.000); *I*^2^ = 81.58%	
More than 24 weeks	1	1.890	1.118 to 2.662	*NA*	

Note. ** < 0.01.

## Data Availability

The data presented in the study are available upon request from the corresponding author.

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
