# Peer review of "The Effectiveness of Physical Activity Interventions on Depression in Korea: A Systematic Review and Meta-Analysis"

_healthcare, 2022, doi:10.3390/healthcare10101886_

Round 1

Reviewer 1 Report

This systematic review and meta-analysis aimed to investigate the effectiveness of physical activity on depression in Korea. In addition, the current review aimed to provide the effect of physical activity on depression according to type, duration, intensity and frequency of exercise.

This research topic is original and provide a significant contribution to the current literature. Different line of evidence has been reported and described Korea as a high prevalence country of suicide. Moreover, the depression in Korea is highly prevalence for different reasons. Physical activity provides different benefits with less side effect. Previous clinical studies encouraging and recommended people to practice physical activity regularly. Thus, this review is providing important evidence regarding the effect of physical activity on depression. 

The main strength point of this paper is the aim, its specific for one country. In addition, investigating the effect of exercise parameters on depression is provide a new evidence in this filed.

The exclusion criteria is not clear, I suggest the authors to include it.

Why the authors limiting their search only in Korean data based?, I understood the language barriers, I don’t think there are not any RCTs published in English in the other databases. I suggest to explain this point and to address it in the limitations section. I suggest also to provide a clear and detailed search strategy as appendix.

The conclusions consistent with the evidence and arguments presented.

The references are appropriate, they are appropriate and covered everything.

The authors stated in figure 1 that they excluded 205 studies? Please provide the reasons of exclusion? Table 2, the authors used SMD, please provide using SMD in the statistical analysis section.

This systematic review and meta-analysis investigated the effectiveness of physical activity on depression in Korea. Indeed, physical activity provides many beneficial effects, especially in mental disorders such as depression. The current review provides new evidence and valuable results that need to be taken into consideration in clinical practice. Minor revisions are needed and listed below:

Page 1, lines 38-41. Please cite this sentence.

The methods, results, and discussion sections are written well.

I recommend accepting this review as it is.

Reviewer 2 Report

The authors highlighted the importance of PA on depression through a meta-analysis. I have a strong agreement on its effect and the authors well described the issues and aim of the study. I think this manuscript is ready to be published in Healthcare with the minor revision as follows.

In Methods, as far as I understand, authors need to choose one of the models, a fixed or a random model. I could be wrong, but it should be clear (Line 111). Also, I think authors do not have to mention the regression-based effect because they used only the mean-difference model (Line 140). Please check Line 168 in this regard as well.

In table 1, I could not understand what ‘mage’ indicated. Please specify it.

Maybe, there are some additional limitations. In depression measurement, I guess the 18 studies did not use a same scale. Although the meta-analysis standardized the scores, it could be a potential risk to defining depression. Each study may have different constructs of depression. In the subgroup analysis, some group has a very low number of studies, which may affect the validity of the interpretation.

Reviewer 3 Report

I thank the authors for submitting their manuscript. However, some adjustments are necessary:

- Change elderly people x older people

- Check the keywords, if possible, thar are MeSH terms.

- The FITT recommendation refers of the principles of training, I think it is more appropriate to refer to it, since one of the aims includes period.

- In methodology, they mus follow the latest PRISMA guidelines (Page et al., 2020).

- According to the new PRSIMA recommendations, the year restriction is not used.

- In my opinion, there is a high bias in using only local databases, as they may not include valuable studies with the Korean population that are found in the main databases such as WoS, Scopus or PubMed. Therefore, it would be better to include the key term Korea OR Korean to improve the collection of retrieval.

- The meta-abalysis is solid. However, the review lacks analyzes of the methodological quality of the studies (PEDRO, TESTEX, others) and certainty of evidence (GRADE) to draw definitive conclusions.

- The results and discussion seem appropriate. However, by not knowing the analysis of the methodological quality of the studies as well as the certainty of evidence, it si not possible to endorose them.

Round 2

Reviewer 3 Report

I appreciate the adjustments made by the authors, wich have substantially improved the manuscript. However, it is necessary to review the citation system (there are some in superscript or in APA format). In addition, I did not find the citation of Page et al (2020) or another referring to the PRISMA standards adjusted in 2020, but they still indicate the of 2009, wich is no longer used.

Author Response

I appreciate the adjustments made by the authors, wich have substantially improved the manuscript. However, it is necessary to review the citation system (there are some in superscript or in APA format). In addition, I did not find the citation of Page et al (2020) or another referring to the PRISMA standards adjusted in 2020, but they still indicate the of 2009, wich is no longer used.

Thank you for your feedback. We have checked the citation system throughout the manuscript and  used page et al. (2021) reference [31] for the PRISMA standards.

Page, M.J.; McKenzie, J.E.; Bossuyt, P.M. et al.The PRISMA 2020 statement: an updated guideline for reporting systematic reviews. Syst Rev. 2021, 10, 89.